# Development and validation of a practical score to predict 3-year distant metastatic free survival in nasopharyngeal carcinoma incorporating the number of lymph node regions

Thitiporn Jaruthien[1,2], Chawalit Lertbutsayanukul[1], Nutchawan Jittapiromsak[3], Aniwat Sriyook[3], Prapasri Kulalert[2], Pichaya Tantiyavarong[2], Sarin Kitpanit[1], Danita Kannarunimit[1], Chakkapong Chakkabat[1], Anussara Prayongrat[1] *

1 Division of Radiation Oncology, Department of Radiology, Faculty of Medicine, Chulalongkorn University, King Chulalongkorn Memorial Hospital, Thai Red Cross Society, Pathumwan, Bangkok, Thailand,
2 Department of Clinical Epidemiology, Faculty of Medicine, Thammasat University, Pathum Thani, Thailand,
3 Division of Diagnostic Radiology, Department of Radiology, Faculty of Medicine, Chulalongkorn University, King Chulalongkorn Memorial Hospital, Thai Red Cross Society, Pathumwan, Bangkok, Thailand

* anussara.p@chula.ac.th

**Data Availability Statement:** All relevant data are within the manuscript and its Supporting information files.

## Abstract

### Introduction

The improvement in diagnosis and treatment for nasopharyngeal carcinoma (NPC) has shifted the pattern of failure toward distant metastasis. This study aimed to develop a simplified prognostic scoring model to predict distant metastatic free survival (DMFS) for NPC patients.

### Materials and methods

Patients with non-metastatic NPC were identified from a retrospective cohort diagnosed between 2010 and 2018. Flexible parametric survival analysis was used to identify potential predictors for DMFS and establish a scoring model. The prognostic accuracy between the 8th AJCC system and the scoring model was compared using Harrell's C-index.

### Results

Of the total 393 patients, the median follow-up time was 85 months. The 3-year DMFS rate was 83.3%. Gender, T-stage, pre-EBV (cut-off 2300 copies/ml), and the number of metastatic lymph node regions were identified as independent risk factors for distant metastasis and were included in the final scoring model. Our established model achieved a high C-index in predicting DMFS (0.79) and was well-calibrated. The score divided patients into two categories: low-risk (score 0–4) and high-risk (score 5–7), corresponding with the predicted 3-year DMFS of 96% and 64.5%, respectively.

**Funding:** The author(s) received no specific funding for this work.

**Competing interests:** The authors have declared that no competing interests exist.

## Conclusions

A feasible and applicative prognostic score was established and validated to discriminate NPC patients into low- and high-risk groups.

## Introduction

Nasopharyngeal carcinoma (NPC) is one of the most common head and neck cancers, with an estimated 130,000 new cases worldwide in 2020. It has a distinct epidemiological feature, as more than 80% of the cases occur in Asia, particularly in southern China and Southeast Asia [1]. Chemoradiotherapy is a mainstay treatment in NPC for stages II-IV, while in stage I, radiotherapy (RT) alone is the standard of care with good efficacy. Improvements in imaging at diagnosis and radiation techniques have shifted the pattern of recurrence from locoregional recurrence toward distant metastasis (DM). After distant progression occurs, the prognosis for this patient group remains poor, with a 5-year survival rate of less than 30% [2, 3]. Therefore, precise risk estimation for DM is essential for optimizing treatment.

Currently, the 8[th] edition American Joint Committee on Cancer (AJCC) staging system, which uses a tumor, node, metastasis (TNM), has been widely accepted for risk stratification and treatment decisions [4]. However, a recent study found that using only TNM staging had limitations in portraying the risk of DM consistently within each stage [5]. Patients within the same TNM stage receiving similar treatments exhibited varying outcomes. Recently, an increasing number of predictive models have been developed to assist physicians in tailoring personalized treatment based on individual risk factors. Previous models for predicting distant metastatic-free survival (DMFS) were based on sophisticated approaches, such as gene expression, radiomic features, or positron emission tomography–computed tomography [6–9]. However, practical models to assess the risk of DM have been limited.

Recognizing additional prognostic factors and developing precise tools to predict the risk of DM are essential. Several studies have demonstrated that certain baseline characteristics, such as male sex and advanced age, increase the risk of DM [10, 11]. Currently, it is widely accepted that the Epstein-Barr virus (EBV) plays a pivotal role in initiating, developing, and progressing of disease. Numerous studies have indicated that the circulating plasma EBV DNA concentration can predict patient prognosis in the early stage of NPC management [12–14]. Moreover, certain distinct characteristics of lymph nodes from magnetic resonance imaging (MRI), such as the size, volume, extracapsular extension, nodal necrosis, and the number of metastatic lymph node regions were found to be independent predictors for DM [15–18]. All these variables are easily obtained from blood examinations and imaging modalities, routinely used in the diagnosis and treatment for NPC patients. To the best of our knowledge, limited models have incorporated clinical variables, hematological biomarkers, and imaging features to predict the risk of distant metastasis [19–21].

The aim of this study was to develop a predictive model using easily obtainable prognostic variables at the time before starting the treatment, to help predict DMFS. This simplified model could routinely be employed in clinical settings to assist physicians in promptly selecting the individualized treatment.

## Materials and methods

A prognostic predictive model was developed using data collected through retrospective medical record reviews. The data were accessed between May 27, 2021, and July 25, 2022, after

approval from the ethics committee, using a study number without the patient's ID, HN, or name to protect patient confidentiality. Patients with non-metastatic NPC were identified from a retrospective cohort diagnosed between 2010 and 2018. The inclusion criteria were as follows: (a) newly diagnosed biopsy-proven squamous cell carcinoma of the nasopharynx, (b) stage II-IVa, (c) age > 18 years old, (d) no evidence of distant metastasis proven by chest X-ray, ultrasonography of the liver, and Tc99m-methylene diphosphonate bone scan, (e) underwent MRI of the head and neck before starting the treatment, (f) treated with curative intent, (g) radiotherapy treatment using intensity-modulated radiation therapy (IMRT) or volumetric modulated arc therapy (VMAT). Patients were excluded if they had incomplete treatment, a follow-up time of less than 2 years, induction chemotherapy, or other malignancies that could interfere with the treatment outcome. This study was approved by the Institutional Review Board of the Faculty of Medicine, Chulalongkorn University, Bangkok, Thailand (IRB no. 768/63) and was conducted following the Declaration of Helsinki. Due to the retrospective analysis of the results, a waiver of informed consent was obtained.

Demographic, tumor characteristics, and baseline laboratory data were obtained from electronic medical records. Pretreatment MRI was reviewed by experienced head and neck radiologists to determine the TNM classification according to the eighth edition of the AJCC/UICC staging system and the lymph node characteristics including the number of metastatic lymph node regions (LNR), necrotic features, and extracapsular extension (ECE).

The nodal level classification was mapped following the eighth edition of the AJCC staging system [4]. Assessed regions included bilateral IA, IB, IIA, IIB, III, IV, VA, VB, VI, and VII. For retropharyngeal LN (RP), bilateral RP was considered as one unit when counting the number of LNR. LNs located on the border of neighboring levels were recorded as involving both regions. More details on diagnostic criteria for metastatic lymph nodes such as central necrosis, ECE, and a summary of the imaging-based nodal level classification can be found in the S1 Appendix.

Plasma EBV DNA levels were collected before treatment (pre-EBV). The plasma DNA samples were quantified for EBV DNA using an RTQ-PCR system targeting the BamHI-W fragment region of the EBV genome. A plasma EBV DNA concentration of < 316 copies/ml was defined as an undetectable level in our institution.

## Treatment

All patients received concurrent chemoradiotherapy, with or without adjuvant chemotherapy. The chemotherapy regimen consisted of platinum-based chemotherapy administered weekly or tri-weekly, concurrently with definitive radiotherapy at a dose of 70 Gy in 33–35 fractions. Adjuvant chemotherapy regimens included cisplatin/5-fluorouracil or carboplatin/5-fluorouracil given at 4-week intervals for 3 cycles. Further details of the treatment can be found in the S1 Appendix.

## Follow-up

Patients were monitored weekly during chemoradiation, before each cycle of adjuvant chemotherapy, and 1 month after completing treatment. Fiberoptic nasopharyngeal examinations, along with CT or MRI scans of the nasopharynx, were conducted 3 months after completing chemoradiation to assess tumor response. The patients were evaluated every 3–6 months during the first 3 years, every 6 months from the fourth to the fifth year, and annually thereafter. Each follow-up visit included a physical examination, endoscopic examination, and blood tests. In cases where there was clinical suspicion of locoregional recurrence or distant metastasis, additional imaging and/or tissue biopsies were performed to confirm disease progression.

## Predictors selection

Candidate predictors included age, sex, T and N staging (8th AJCC), pretreatment EBV DNA level, MRI-based number of LNR, and characteristics. The selection of predictors was based on their availability at the time of prediction and previously reported models for predicting DMFS in NPC. The full model included a total of 8 predictors, with age and LNR categorized at acceptable cutoff values from previous studies [11, 16]. Based on our previous study, the established cut-off value for the pre-treatment EBV level was 2,300 copies/ml [14, 22].

## Statistical analysis

Analysis was carried out using Stata/SE 18.0 (StataCorp, Texas, USA). Continuous variables were reported as mean (with standard deviation [SD]). Categorical variables were presented as counts and percentages. DMFS was measured from the date of the start of treatment until the date of proven metastasis. Overall survival (OS) was measured from the date of the start of treatment until the date of death (any cause). Patients who were still alive were censored at their last follow-up date. Patients without any endpoints were censored on July 25, 2022. DMFS and OS were calculated using the Kaplan-Meier method and the differences were compared using the log-rank test.

A flexible parametric survival model, developed by Royston and Parmar in 2002, was used to derive the prognostic model via the stpm2 STATA package. The advantage of this model over the Cox regression model is its ability to estimate the baseline cumulation hazard function, which allows more accurate prediction. Parametric survival models, such as the exponential and Weibull models, also make strong assumptions about the shape of the baseline hazard function. Therefore, a flexible parametric survival model is more flexible to not misspecified baseline hazard shapes and provides more accurate estimations. In our model, we opted for a cumulative hazard scale featuring two degrees of freedom after considering the criteria of the lowest Akaike information criterion (AIC) and Bayesian information criterion (BIC) values. The proportional hazard assumption was tested using Schoenfeld residuals before deriving the model.

Eight potential predictors were included in the multivariable flexible parametric model. Variables selection was conducted using stepwise backward elimination with a significance threshold of P-value less than 0.05. Model discriminative performance was measured using Harrell's c-index and was compared with the 8[th] AJCC system by applying the T and N staging in the flexible parametric model. We assessed the calibration of the derived model by using calibration plots. We performed internal validation using a bootstrapping procedure with 500 bootstrap samples. This procedure quantified the optimism of the developed model.

To generate the clinical prediction score, the coefficients of all predictors were weighted by dividing the lowest coefficient, and any result equal to or greater than 0.5 was rounded up to the nearest integer. For clinical implications, we categorized the prediction score into two risk groups: low-risk and high-risk groups using the 80% cut-off of 3-year DMFS. The analysis was conducted using the complete-case method without data imputation, as missing data for all variables was less than 5%.

According to the Chinese Society of Clinical Oncology (CSCO) and the American Society of Clinical Oncology (ASCO) guidelines [23], for patients with locoregionally advanced NPC stage III-IV (accepted T3N0), induction chemotherapy is recommended in addition to concurrent chemoradiotherapy (CCRT) or CCRT plus adjuvant chemotherapy due to distinctly poor survival outcomes. For stage II to early stage III (T3N0), which comprise heterogeneous groups of patients, there is a need to identify the low-risk cohort for de-intensified treatment and the high-risk cohort for treatment intensification. For example, for patients with T1-2N0-

1 and T3N0 NPC, induction/adjuvant chemotherapy is not routinely recommended but may be offered if there are adverse features, such as bulky tumor volumes or high EBV DNA copy number. Therefore, we performed a post-hoc subgroup analysis for stage II to early stage III (T3N0) NPC. The aim was to identify the low-risk cohort for de-intensified treatment and the high-risk cohort for treatment intensification using the proposed score model.

## Results

Between January 2010 and December 2018, 547 patients met our eligibility criteria. After excluding 147 patients according to the exclusion criteria and excluding 7 patients for missing pre-treatment plasma EBV levels, 393 patients were included for model development (Fig 1). The mean age was 50 years, with males predominating. Patient characteristics are outlined in Table 1. The median follow-up time was 85 months. A total of 71 patients developed distant metastasis (18%). A total of 110 patients died (28%). The 3- and 5-year DMFS rates were 83.3% and 81.2%, respectively. The overall survival rates at 3 and 5 years were 84.5% and 77.2%, respectively.

### Potential predictors

From the univariable flexible parametric survival analysis, eight predictors were identified as candidate predictors of DMFS: aged ≥ 60 years, male gender, T stage, N stage, pre-treatment EBV level ≥ 2,300 copies/mL, number of LNR, the presence of necrotic LN and the presence of ECE. All candidate predictors listed in Table 2 were included in the full multivariable flexible parametric survival analysis. No statistical evidence of a violation of the proportional hazard assumption was found in the Schoenfeld residuals test (P = 0.43). The reduced model was generated through stepwise backward elimination based on a P value < 0.05. The four final

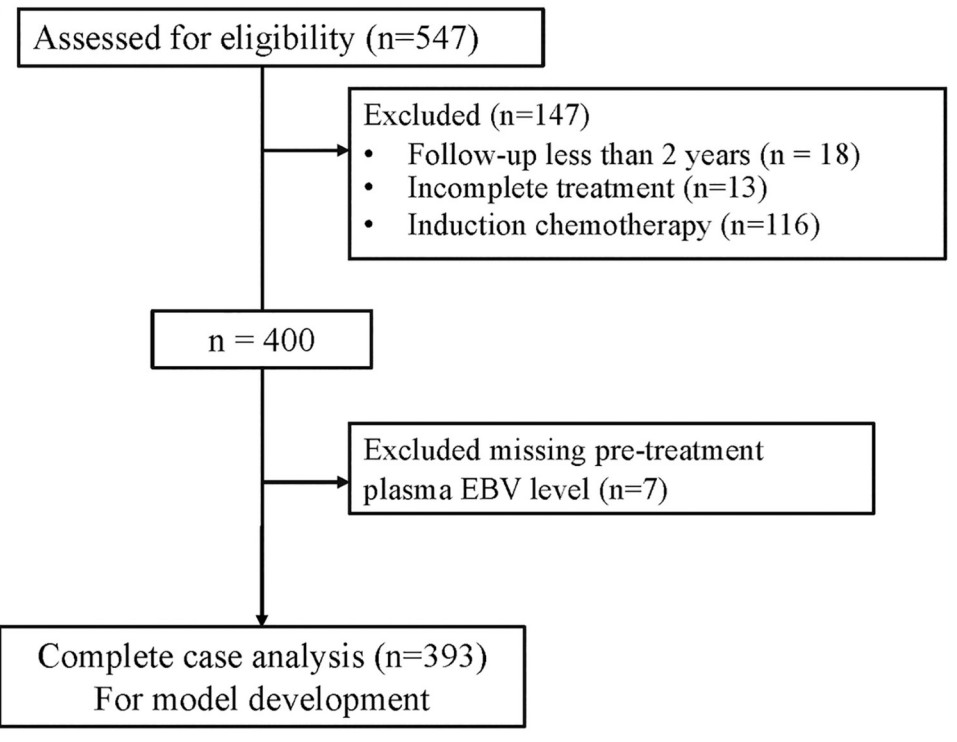

**Fig 1. Study flow diagram.**

**Table 1. Baseline characteristics of patients.**

| Characteristics | Total N = 393 |
|---|---|
| | N (%) |
| **Mean age (SD), years** | 50 (12) |
| <60 | 304 (77.4) |
| ≥60 | 89 (22.6) |
| **Sex** | |
| Female | 108 (27.5) |
| Male | 285 (72.5) |
| **Histologic type** | |
| Nonkeratinizing SCCA | 72 (18.3) |
| Undifferentiated SCCA | 320 (81.4) |
| Basaloid SCCA | 1 (0.3) |
| **T category (AJCC 8$^{th}$)** | |
| T1 | 105 (26.7) |
| T2 | 80 (20.4) |
| T3 | 133 (33.8) |
| T4 | 75 (19.1) |
| **N category (AJCC 8$^{th}$)** | |
| N0 | 20 (5.1) |
| N1 | 214 (54.5) |
| N2 | 98 (24.9) |
| N3 | 61 (15.5) |
| **Stage group (AJCC 8$^{th}$)** | |
| II | 118 (30) |
| III | 150 (38.2) |
| IVA | 125 (31.8) |
| **Pre-treatment plasma EBV DNA level (copies/mL)** | |
| <2300 or undetectable | 215 (54.7) |
| ≥ 2300 | 178 (45.3) |
| **Number of LN region (LNR)** | |
| 0–1 | 113 (28.8) |
| 2–6 | 225 (57.2) |
| 7–13 | 55 (14) |
| **Necrotic LN** | |
| No | 223 (56.7) |
| Yes | 170 (43.3) |
| **Extracapsular extension (ECE)** | |
| No | 360 (91.6) |
| Yes | 33 (8.4) |
| **Concurrent chemotherapy** | |
| Weekly cisplatin | 257 (65.4) |
| Cisplatin tri-weekly | 79 (20.1) |
| Weekly carboplatin | 35 (8.9) |
| Carboplatin tri-weekly | 12 (3.1) |
| Weekly carboplatin/paclitaxel | 2 (0.5) |
| Missing | 8 (2) |
| **Cumulative cisplatin dose** | |
| ≥200 mg/m$^2$ | 291 (86.6) |

*(Continued)*

**Table 1.** (Continued)

| Characteristics | Total N = 393 |
|---|---|
| | N (%) |
| <200 mg/m$^2$ | 38 (11.3) |
| Unknown doses | 7 (2.1) |
| **Adjuvant chemotherapy** | |
| None | 64 (16.3) |
| 1 cycle | 26 (6.6) |
| 2 cycles | 28 (7.1) |
| 3 cycles | 257 (65.4) |
| Unknown number of cycles | 9 (2.3) |
| Missing | 9 (2.3) |

Abbreviations: SD; standard deviation, SCCA; squamous cell carcinoma, AJCC; American Joint Committee on Cancer, EBV; Epstein-Barr virus

**Table 2. Estimated hazard ratios in the univariable and multivariable flexible parametric regression models.**

| Predictors | Univariable model | | | Multivariable model | | |
|---|---|---|---|---|---|---|
| | HR | 95% CI | P- value | HR | 95% CI | P- value |
| **Age <60** | 1 | | | | | |
| **Age ≥ 60** | 1.61 | 0.96–2.67 | 0.069 | | | |
| **Female** | 1 | | | 1 | | |
| **Male** | 2.84 | 1.41–5.72 | 0.003 | 2.51 | 1.24–5.07 | 0.01 |
| **T1** | 1 | | | 1 | | |
| **T2** | 2.16 | 0.93–4.98 | 0.072 | 2.02 | 0.87–4.69 | 0.103 |
| **T3** | 2.88 | 1.36–6.11 | 0.006 | 2.67 | 1.25–5.69 | 0.011 |
| **T4** | 3.85 | 1.75–8.47 | 0.001 | 2.91 | 1.32–6.42 | 0.008 |
| **N 0–1** | 1 | | | | | |
| **N2** | 2.46 | 1.41–4.28 | 0.001 | | | |
| **N3** | 4.02 | 2.25–7.18 | <0.001 | | | |
| **Pre-treatment EBV** | | | | | | |
| **<2,300** | 1 | | | 1 | | |
| **≥2,300** | 3.2 | 1.93–5.29 | <0.001 | 1.90 | 1.12–3.24 | 0.018 |
| **No of LNR** | | | | | | |
| **0–1** | 1 | | | 1 | | |
| **2–6** | 4.48 | 1.77–11.35 | 0.002 | 3.99 | 1.55–10.25 | 0.004 |
| **7–13** | 14.71 | 5.65–38.34 | <0.001 | 9.36 | 3.46–25.30 | <0.001 |
| **Presence of LN Necrosis** | | | | | | |
| **No** | 1 | | | | | |
| **Yes** | 2.40 | 1.49–3.88 | <0.001 | | | |
| **Presence of ECE** | | | | | | |
| **No** | 1 | | | | | |
| **Yes** | 2.63 | 1.41–4.88 | 0.002 | | | |

Abbreviations: EBV; Epstein-Barr virus, LN; lymph node, LNR; number of lymph node region, ECE; radiologic gross extracapsular extension

**Table 3. Best multivariable clinical predictors, hazard ratio (HR), 95% confidential interval (CI), regression beta coefficient (β), and assigned item score.**

| Predictors | Multivariable model | | | β *coeff* | *Score* |
|---|---|---|---|---|---|
| | **HR** | **95% CI** | ***P*- value** | | |
| **Female** | 1 | | | | 0 |
| **Male** | 2.51 | 1.24–5.07 | 0.01 | 0.920 | 1 |
| **T1** | 1 | | | | 0 |
| **T2** | 2.02 | 0.87–4.69 | 0.103 | 0.701 | 1 |
| **T3** | 2.67 | 1.25–5.69 | 0.011 | 0.980 | 2 |
| **T4** | 2.91 | 1.32–6.42 | 0.008 | 1.068 | 2 |
| **EBV pre-treatment** | | | | | |
| **<2,300** | 1 | | | | 0 |
| **≥2,300** | 1.90 | 1.12–3.24 | 0.018 | 0.642 | 1 |
| **No of LNR** | | | | | |
| **0–1** | 1 | | | | 0 |
| **2–6** | 3.99 | 1.55–10.25 | 0.004 | 1.384 | 2 |
| **7–13** | 9.36 | 3.46–25.30 | <0.001 | 2.236 | 3 |

Abbreviations: EBV; Epstein-Barr virus, LNR; number of lymph node region

predictors include male gender, T stage, pre-treatment EBV level, and number of LNR. The estimated beta coefficients and their 95% confidence intervals are shown in Table 3.

## Clinical prediction score

We used the lowest beta-coefficient, 0.642, as a dominator, and assigned weighted scores: 1 for male gender, T2 stage, and pre-treatment Epstein-Barr virus (EBV) level ≥2,300 copies/mL; 2 for T3 or T4 stage, and the number of lymph node regions (LNR) in the range of 2–6 regions; and 3 for the number of LNR in the range of 7–13 regions (Table 3). The total score ranged from 0 to 7. The cut-off value for the risk score, distinguishing between low-risk and high-risk patients, was set at 5 using the 80% cut-off of 3-year DMFS (S1 Fig). The scores were divided into two categories: low-risk for DMFS (score 0–4) and high-risk for DMFS (score 5–7). The predicted 3-year DMFS for low-risk and high-risk groups were 96% and 64.5%. The predicted 3-year OS for low-risk and high-risk groups were 94.8% and 70.1%. The Kaplan-Meier curves with 95% CIs of 2 risk groups of DMFS and OS are shown in Fig 2 The log-rank test of both graphs yielded a P-value of < 0.001. We also present our final model as a nomogram to predict 3-year DMFS, providing an option for clinicians to use (S2 Fig).

## Model discrimination and calibration

For the measure of discrimination performance, the Harrell C-statistic for the final model and scoring model were 0.79 and 0.78, respectively. The calibration of the final model was visualized with a calibration plot (Fig 3), demonstrating that the prognostic model was well-calibrated.

## Internal validation

Internal validation of the derived prognostic model was performed via a bootstrap resampling method with 500 replicates. The C-statistics of the developed and validated models were 0.79 and 0.78. When comparing the predictive accuracy for DMFS between the derived model and

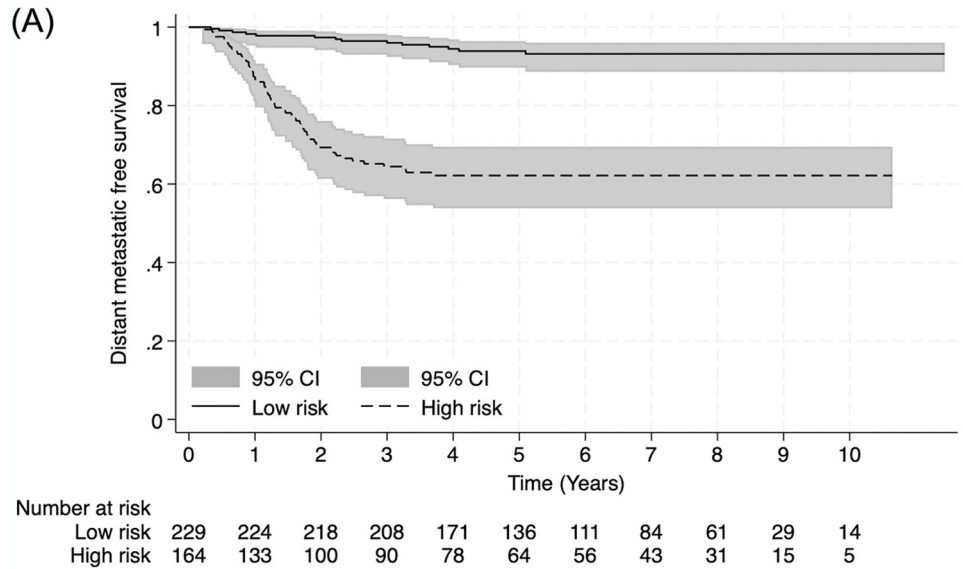

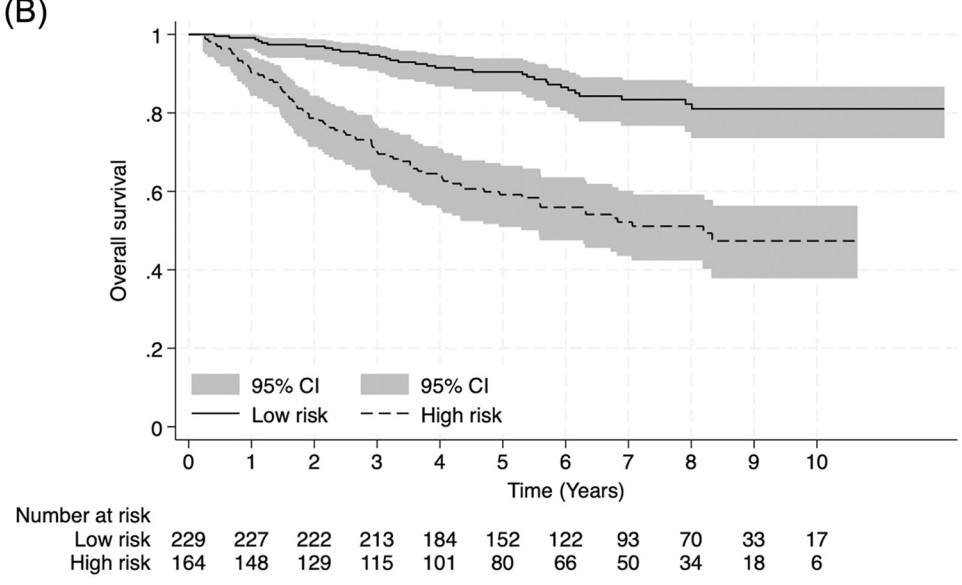

**Fig 2. The Kaplan-Meier curve with 95% CIs of 2 risk groups of DMFS (A) and OS (B).**

the 8th AJCC staging systems, the derived model demonstrated superior accuracy. The c-index of the model was higher than that of the 8th edition of the AJCC staging system (0.79 vs. 0.70).

## Post-hoc subgroup analysis of stage II to early stage III

The subgroup of patients with stages T1-2N0-1 and T3N0 were identified and applied with the proposed score model. There were 124 patients in this group composed of scores 0 to 5. The high-risk group with a score of 5 had significantly worse DMFS and OS compared to the low-risk group (Fig 4).

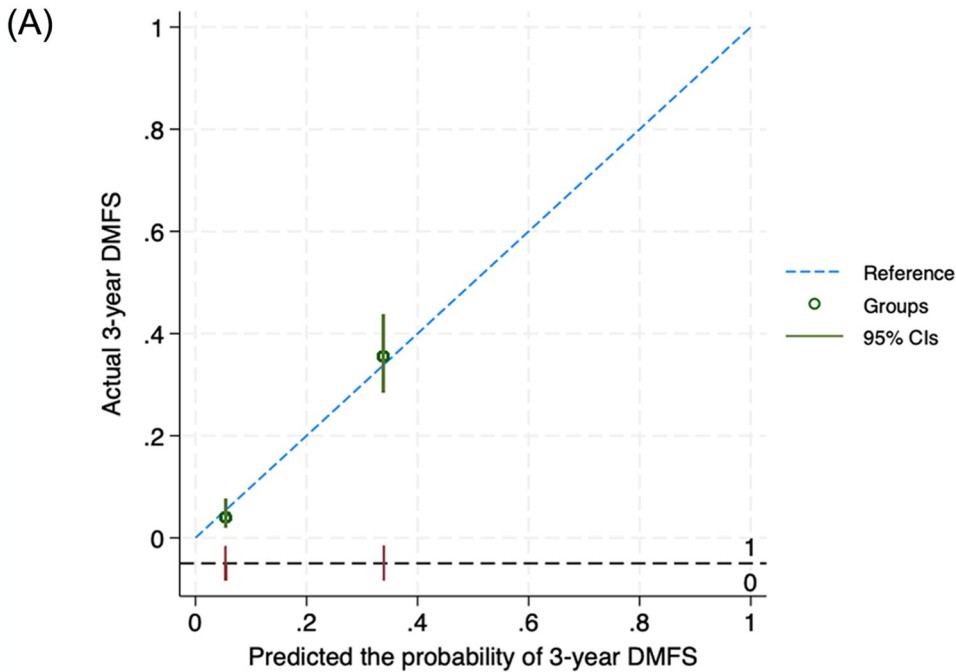

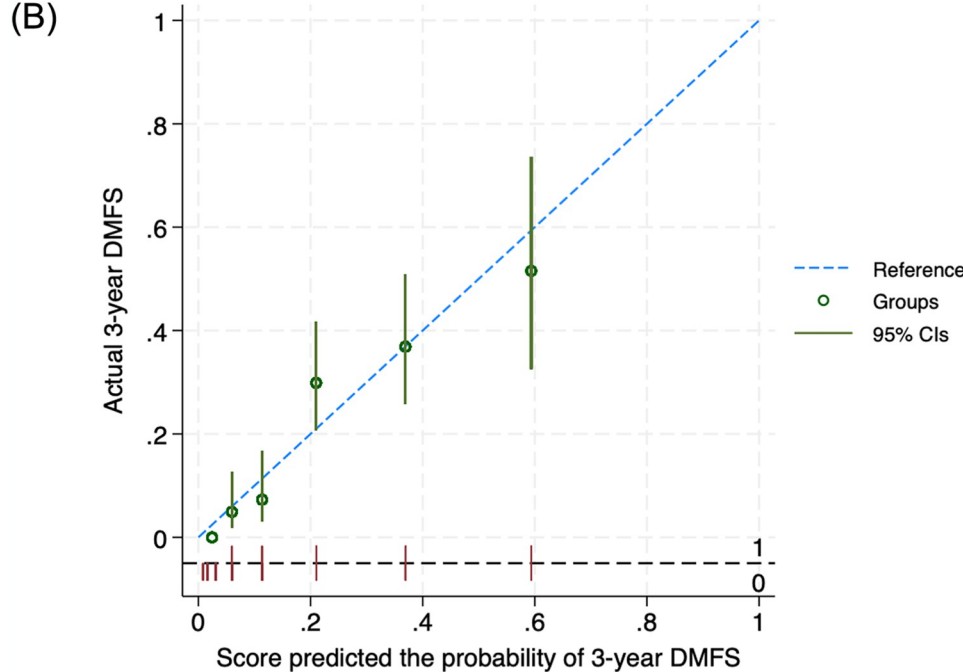

**Fig 3. Calibration plots compare the model-predicted probability of 3-year DMFS and the observed outcomes against one another within each of the risk groups (A) and score (B).**

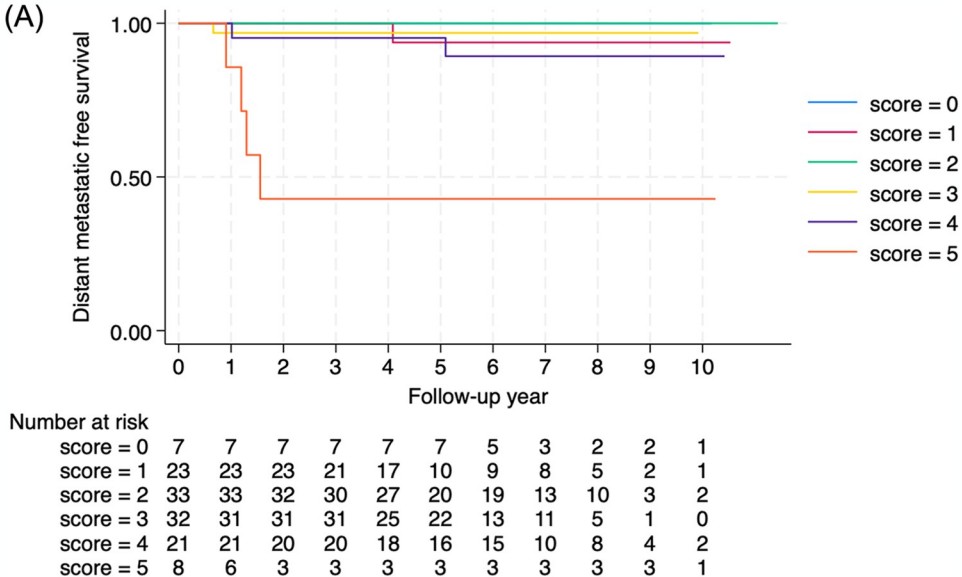

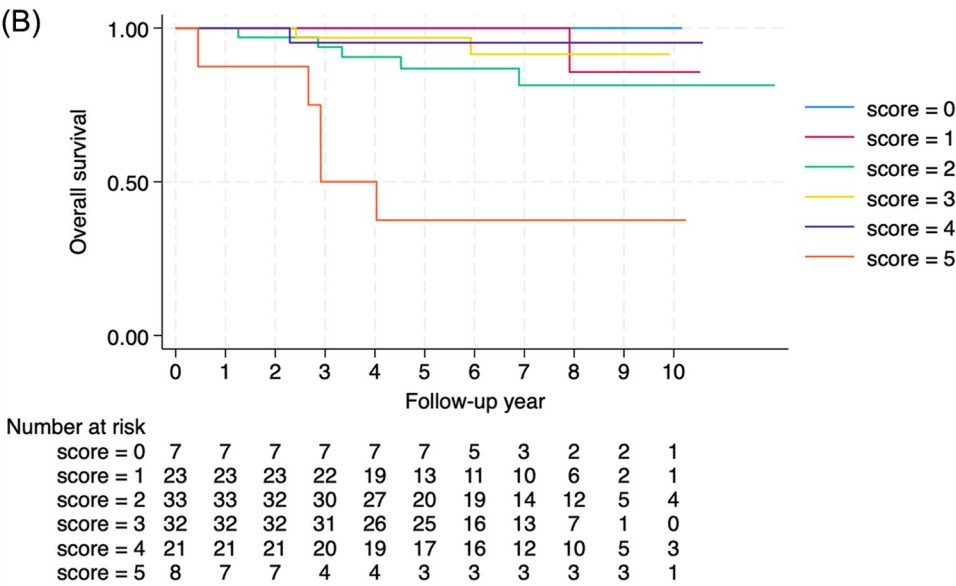

**Fig 4. The Kaplan-Meier curve of DMFS (A) and OS (B) for patients with T1-2N0-1 and T3N0 NPC according to the prediction score.**

## Discussion

In this study, we developed a prognostic score for predicting DMFS in patients with NPC. Our model incorporates simple clinical characteristics, hematological biomarkers, and LN characteristics from imaging. The scoring model was designed for simplicity, user-friendliness, and utilization of readily available parameters. It demonstrated improved prognostic accuracy compared to the current staging system, with a Harrell's C-statistic of 0.79 versus 0.70. This

indicates that the model recognizes additional prognostic factors and provides a more precise tool to predict the risk of distant metastasis (DM).

Compared to other previous models, our model is the first to incorporate LNR into the prognostic score. Quantitative lymph node burden has been demonstrated to be a significant prognostic factor in various malignancies such as breast cancer, colorectal cancer, and squamous head and neck cancers. For instance, the number of metastatic LNs is a promising novel predictor of survival with demonstrated superiority to the 8[th] edition AJCC N classification in many squamous head and neck cancers [24, 25]. For NPC, pathological quantification of LNs is unavailable. Therefore, the current N classification system is based on two-categorical nodal laterality, level, and size. The 8[th] AJCC N classification system does have limitations; for instance, patients with extensive metastatic LNs could be staged the same as those with single LN despite their much poorer prognosis. In the study by Zhou et al. [16], they reported 5-year DMFS rates for LNR 0–1, 2–6, and ≥ 7 as 97%, 86.7%, and 69.7%, respectively. Their findings demonstrated an improved discrimination capability for DMFS compared with the 8th edition of AJCC N classification. Xie et al. [20] developed a nomogram incorporating nodal numbers which might be too laborious to apply in real-world settings. The difficulty arose from the challenge of counting nodal numbers accurately from imaging, particularly when two or more nodes coalesced. On the other hand, LNR was routinely reported by radiologists in the imaging report without requiring additional workload.

Moreover, it might assist in identifying low-risk and high-risk NPC candidates who could benefit from de-intensified or more intensified treatment. For example, our post-hoc subgroup analysis for patients with T1-2N0-1 and T3N0 NPC comprises heterogeneous groups of patients. According to CSCO and ASCO Guideline, induction/adjuvant chemotherapy is not routinely recommended for this group but may be offered if there are adverse features, such as bulky tumor volumes or high EBV DNA copy numbers. Therefore, when applying the score model to the subgroup, the high-risk group with a score of 5 had significantly worse DMFS and OS compared to the low-risk group, suggesting intensified treatment for this group. This scoring model is the first predictive model for DMFS in NPC that uses flexible parametric survival analysis, which surpasses Cox regression in its ability to estimate the baseline cumulative hazard function, enabling more accurate survival predictions.

Nevertheless, the present study had several limitations. Firstly, being conducted in a single institution population with a relatively small sample size, external validation with a larger cohort should be warranted. Secondly, we used the pre-EBV cut-off of 2,300 copies/ml which was different from studies from the Chinese population [12, 13]. However, since there is no standard pre-EBV cut-off value, our previous report suggested that this cut-off level was optimal for predicting DMFS [22]. Thirdly, since we aimed to develop a model for pre-treatment prediction using uniformly treated patients for the accuracy of the prediction model, our study did not include patients who received induction chemotherapy, which might have a higher risk for distant metastasis and could introduce bias as confounding by indication. However, induction chemotherapy was not a standard treatment during the study period and was only adopted a few years before the study concluded. Validation with this group of patients is warranted.

## Conclusion

We established and validated a simplified score model to predict DMFS in NPC patients, incorporating gender, T-stage, pre-EBV level, and number of LNR. This model can support physicians in decision-making for optimal management and exhibits higher predictive power

compared to the traditional TNM staging system, especially in subgroups of patients with stage II to early stage III.

## Supporting information

**S1 Appendix. Criteria for diagnosis of nodal metastasis and lymph node characteristics in magnetic resonance imaging, criteria for lymph node regions (LRN), and details of the treatment.**
(DOCX)

**S1 Data. Patients' information.**
(PDF)

**S1 Fig. The Kaplan-Meier curve of DMFS according to the prediction score.**
(TIF)

**S2 Fig. The nomogram to predict 3 year DMFS.**
(TIF)

## Author Contributions

**Conceptualization:** Thitiporn Jaruthien, Chawalit Lertbutsayanukul, Anussara Prayongrat.

**Data curation:** Thitiporn Jaruthien, Nutchawan Jittapiromsak, Aniwat Sriyook.

**Formal analysis:** Pichaya Tantiyavarong.

**Investigation:** Thitiporn Jaruthien, Sarin Kitpanit, Danita Kannarunimit, Chakkapong Chakkabat.

**Methodology:** Thitiporn Jaruthien, Prapasri Kulalert, Pichaya Tantiyavarong.

**Supervision:** Chawalit Lertbutsayanukul, Prapasri Kulalert, Anussara Prayongrat.

**Validation:** Thitiporn Jaruthien.

**Writing – original draft:** Thitiporn Jaruthien.

**Writing – review & editing:** Chawalit Lertbutsayanukul, Nutchawan Jittapiromsak, Aniwat Sriyook, Prapasri Kulalert, Pichaya Tantiyavarong, Sarin Kitpanit, Danita Kannarunimit, Chakkapong Chakkabat, Anussara Prayongrat.

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
