## [Decision Letter · Decision Letter 0]

23 Jul 2024

PONE-D-24-23002Development and validation of a practical score to predict 3-year distant metastatic free survival in nasopharyngeal carcinoma incorporating the number of lymph node regions.PLOS ONE

Dear Dr. Prayongrat,

Thank you for submitting your manuscript to PLOS ONE. After careful consideration, we feel that it has merit but does not fully meet PLOS ONE’s publication criteria as it currently stands. Therefore, we invite you to submit a revised version of the manuscript that addresses the points raised during the review process.

We look forward to receiving your revised manuscript.

Kind regards,

Lu Zhang

Academic Editor

PLOS ONE

Journal Requirements:

Reviewers' comments:

Reviewer's Responses to Questions

**Comments to the Author**

1. Is the manuscript technically sound, and do the data support the conclusions?

Reviewer #1: Yes

Reviewer #2: Yes

2. Has the statistical analysis been performed appropriately and rigorously? 

Reviewer #1: Yes

Reviewer #2: I Don't Know

3. Have the authors made all data underlying the findings in their manuscript fully available?

Reviewer #1: Yes

Reviewer #2: Yes

4. Is the manuscript presented in an intelligible fashion and written in standard English?

Reviewer #1: Yes

Reviewer #2: Yes

5. Review Comments to the Author

Reviewer #1: The sample size is good for this research work. But this work is confined only with deriving a range of scores. A complete nomogram with such a good sample may give a clear understanding of the predictive powers of the 8 candidates. For improving the writing context, author can consider revising the line 304 of Discussion section.

Reviewer #2: Having analysed this manuscript, I believe this is a relevant and interesting piece for publication.

The authors have described their research in a clear and well elaborated manuscript. The data collection and interpretation seems to have been done rigorously and efficiently, with a sample size appropriate for NPC. The protocol appears well thought out, planned and executed.

The discussion was explained well with ample data to substantiate the results. The analysis of the data appears sound and the topic itself is relevant and likely to spark discussion. While the sample is limited to a single institution, this has been acknowledged by the authors. This paper is also likely to encourage similar studies on a larger scale and at other institutions, maybe across a more diverse patient population, which can further substantiate on the topic and provide more data to possibly bolster this study.

The fact that the model developed by the authors can support clinicians in decision-making, treatment planning, risk estimation and predicting the outcome of patients with Nasopharyngeal carcinoma, especially compared to the TNM staging that is currently predominantly in use, makes it clinically relevant in the field of Head and Neck oncology. This is especially more feasible since the criteria evaluated in the study to predict distant metastasis are investigations that are relatively inexpensive and accessible in most cancer centres such as MRIs, ultrasounds, EBV levels and chest x-rays rather than more sophisticated genetic tests or PET-CT.

I am unable to satisfactorily comment on the statistical analysis as someone who has not had extensive experience in the same, so shall defer the same to fellow reviewers with more expertise.

I have also acknowledged the authors' declaration that all the gathered data was collected retrospectively and anonymously, and cannot be shared publicly because of local ethics committee regulations, and hence is not entirely available in the manuscript.

The language used in the manuscript is clear and grammatically correct. I do not anticipate any further editing to be necessary in this respect.

6. PLOS authors have the option to publish the peer review history of their article (what does this mean?). If published, this will include your full peer review and any attached files.

Reviewer #1: **Yes: **Surega Anbumani

Reviewer #2: No

---

## [Author Response · Author response to Decision Letter 0]

1 Aug 2024

Thank you very much for carefully reviewing our manuscript and giving us very useful comments and the opportunity to revise. We have made changes in the manuscript according to the suggestions as the followings:

Editor

A1: We have checked PLOS ONE's style requirements as you suggested.

2. We note that you have indicated that there are restrictions to data sharing for this study. For studies involving human research participant data or other sensitive data, we encourage authors to share de-identified or anonymized data.

A2: We have provided all the minimal anonymized data sets necessary to replicate our study findings in the supplementary file (S1 Data) and changed the data sharing statement to: “All relevant data are within the manuscript and its supporting information files.”

3. The ORCiD iD has been validated.

Reviewer #1

Q1: The sample size is good for this research work. But this work is confined only with deriving a range of scores. A complete nomogram with such a good sample may give a clear understanding of the predictive powers of the 8 candidates. For improving the writing context, author can consider revising the line 304 of Discussion section.

A1: We appreciate your suggestion. As recommended, we have created a nomogram based on the final predictors and included it in the supplementary file. We initially selected 8 potential candidate predictors, but only the final four predictors were used in our final model We chose not to include the nomogram in the main manuscript to avoid potentially confusing readers when interpreting the simplified score. We have added the following sentence to the result section “We also present our final model as a nomogram to predict 3-year DMFS, providing an option for clinicians to use (S2 Fig).” Please see page 16, lines 229-230.

We have also revised line 304 in the discussion section to enhance clarity.

All the changes are tracked in the revised manuscript with track changes. A clean manuscript (with no track changes) is also provided. We hope that it now meets all the requirements of the journal. We appreciate the opportunity to contribute to the PLOS ONE journal.

Sincerely yours,

Anussara Prayongrat, MD, PhD

Corresponding Author

---

## [Decision Letter · Decision Letter 1]

13 Aug 2024

Development and validation of a practical score to predict 3-year distant metastatic free survival in nasopharyngeal carcinoma incorporating the number of lymph node regions.

PONE-D-24-23002R1

Dear Dr. Anussara Prayongrat,

We’re pleased to inform you that your manuscript has been judged scientifically suitable for publication and will be formally accepted for publication once it meets all outstanding technical requirements.

Kind regards,

Lu Zhang

Academic Editor

PLOS ONE

Additional Editor Comments (optional):

Reviewers' comments:

Reviewer's Responses to Questions

**Comments to the Author**

1. If the authors have adequately addressed your comments raised in a previous round of review and you feel that this manuscript is now acceptable for publication, you may indicate that here to bypass the “Comments to the Author” section, enter your conflict of interest statement in the “Confidential to Editor” section, and submit your "Accept" recommendation.

Reviewer #1: All comments have been addressed

2. Is the manuscript technically sound, and do the data support the conclusions?

Reviewer #1: Yes

3. Has the statistical analysis been performed appropriately and rigorously? 

Reviewer #1: Yes

4. Have the authors made all data underlying the findings in their manuscript fully available?

Reviewer #1: Yes

5. Is the manuscript presented in an intelligible fashion and written in standard English?

Reviewer #1: Yes

6. Review Comments to the Author

Reviewer #1: All of my comments were addressed. And the manuscript looks okay to get published. My final thought is that the author can mention the radiotherapy technique used for treating NPC patients.

7. PLOS authors have the option to publish the peer review history of their article (what does this mean?). If published, this will include your full peer review and any attached files.

Reviewer #1: **Yes: **Surega Anbumani

---

## [Editor Report · Acceptance letter]

16 Aug 2024

PONE-D-24-23002R1 

PLOS ONE

Dear Dr. Prayongrat, 

I'm pleased to inform you that your manuscript has been deemed suitable for publication in PLOS ONE. Congratulations! Your manuscript is now being handed over to our production team.

Kind regards, 

on behalf of

Dr. Lu Zhang 

Academic Editor

PLOS ONE